# The Relationships among Mindfulness, Self-Compassion, and Subjective Well-Being: The Case of Employees in an International Business

**Feng-Hua Yang [1], Shih-Lin Tan [2,\*] and Yuan-Lie Lin [3]**

[1] Department of International Business Management, Da-Yeh University, Changhua County 515006, Taiwan; leonard@mail.dyu.edu.tw
[2] Ph.D. Program in Management, Da-Yeh University, Changhua County 515006, Taiwan
[3] Wende Elementary School, Banqiao District, New Taipei City 22060, Taiwan; tt121@apps.ntpc.edu.tw
[\*] Correspondence: D9933012@cloud.dyu.edu.tw; Tel.: +886-932-319-240

**Abstract:** This study aimed to investigate the relationships among mindfulness, self-compassion, and subjective well-being among employees. The questionnaire research method was used to collect data in this study, and the subjects included employees of Carrefour, an international business in Taiwan. A total of 629 valid questionnaires were used to evaluate the overall structure and analyze the mediating effect with the SPSS 21.0 statistical software. The results showed that mindfulness was positively related to subjective well-being, mindfulness was positively related to self-compassion, and self-compassion was positively related to subjective well-being. It was also found that self-compassion partially mediated the relationship between mindfulness and subjective well-being.

**Keywords:** mindfulness; self-compassion; subjective well-being; workplace friendship; social support

## 1. Introduction

Since December 2019, COVID-19 has been raging all over the world. Most countries have taken strict blockade and isolation measures to prevent the spread of the pandemic, which led to the shutting down of global economic activities, significantly affecting people's lifestyles. In addition, people's economic income decreased under such a harsh environment, making it inevitable for the objective well-being measured by economic income to decrease accordingly.

"Subjective well-being" is an overall evaluation of individuals' lives according to their own defined standards [1]. Based on the perspective of subjective well-being, more than 99% of people in real life make subjective evaluations of their lives. [2]. Thus, subjective well-being can be defined as a subjective overall evaluation of peoples' lives based on the standards defined by the evaluators themselves. It is an individual's evaluation shaped by their life experience. In contrast, objective well-being refers to the individual's functional capability, health status, and socioeconomic status. While the objective environment and individuals' encounters are indeed related to their life happiness, the process and specific circumstances experienced by individuals also affect their feelings [3]. According to the authors of [4], with respect to the individual, "I have nothing to do with the world. I have to do with the Individual, each Individual, or to each Individual".

Furthermore, subjective well-being is based on personal subjective positive and negative emotions and life satisfaction [5]. People will affirm the needs for self-concept, value, self-satisfaction, interpersonal interaction, and social support through social role participation. In this way, individuals can feel a sense of accomplishment and well-being [6]. Psychological well-being is a structure that includes emotional and perceived levels, and its specific connotation includes positive and negative emotions, happiness, life satisfaction, agreement of expectations for and achievement of life goals, physical and mental harmony,

mood, self-esteem, self-efficacy, and personal autonomy [7]. According to the literature, that social support has a positive relationship with subjective well-being. Many studies on the relationship between age and well-being showed that well-being and happiness would decrease with age [8]. With regards to COVID-19 in 2020, happiness has not decreased because of the pandemic, and women's happiness has not decreased. The main reasons may be other experiences, such as quality of life and place quality. This could also signal a feminist social shift [9].

According to relevant studies, subjective well-being is affected by individual traits and actions, such as being grateful [10], obtaining more social support [11–13], good interpersonal relationships [14–17], family structure, and demographics [18–20]. Mindfulness and subjective well-being are related to "satisfaction" and "emotion" affecting each other. However, it is not clear how mindfulness affects subjective well-being. Therefore, it is necessary to explore the relationship between the two variables.

The concept of self-compassion contains three main components, which interact with each other: self-benevolence, human consensus, and mindfulness [21]. It is emphasized that self-compassion is an interconnection rather than separation, implying that even if one is not "better" than others, one can still feel good about oneself. Therefore, this study explores whether self-compassion is the intermediate variable between mindfulness and subjective well-being [22].

Based on the above premises, this study mainly focuses on the relationships among mindfulness, self-compassion, and subjective well-being of employees in Carrefour, an international business. Through the collection, analysis, and induction of relevant literature, quantitative methods will be used to conduct empirical research in order to explore self-compassion, mindfulness, and the subjective well-being of Carrefour employees.

## 2. Theoretical Background

### 2.1. Implication and Theory of Mindfulness

Mindfulness is often defined as the attentive awareness of an individual of current events and their attitude toward accepting and not judging what has happened [23–25]. The original intention of mindfulness is to hold onto hope that people remember the noble path to achieve happiness and eliminate misery [26]. At first, a study defined mindfulness as the ability to concentrate on an individual's current behavior in a flexible mental state and pay attention to new things and contexts [27]. The study of defined mindfulness as recalling and being aware of some facts, among which the most important thing is to be aware of the impermanence of all physical and mental phenomena [28]. Much literature defines mindfulness as the awareness of one's surroundings, presenting the experience of the moment [29]. Furthermore, mindfulness emphasizes dealing with things in life intelligently and avoiding blind and instinctive reactions [30]. It also emphasizes facing the present experience with curiosity, openness, and acceptance [31].

When a person is in a state of mindfulness, they are intentional. At the same time, they flexibly perceive the internal and external stimuli in every moment and observe all stimuli with an open, interested, friendly, and compassionate attitude, whether these stimuli are pleasant, unpleasant, or neutral.

Mindfulness is a multi-dimensional concept. Through exploratory factor analysis, five main mindfulness characteristics were found: (i) attention/action awareness ("I pay attention to sounds, such as clocks ticking, birds chirping, or cars passing "), (ii) observation ("I remain present with sensations and feelings even when they are unpleasant or painful" and "I pay attention to sounds, such as clocks ticking, birds chirping, or cars passing"), (iii) non-judgment ("I tend to evaluate whether my perceptions are right or wrong" and "I think some of my emotions are bad or inappropriate and I shouldn't feel them"), (iv) non-reactiveness to inner experience ("Usually when I have distressing thoughts or images, I step back and am aware of the thought or image without getting taken over by it"), and (v) description ("I'm good at finding the words to describe my feelings" and "My

natural tendency is to put my experiences into words"), establishing the five-dimension mindfulness questionnaire (FFMQ) [32].

Mindfulness implies a person clearly perceiving his inner self and the external world at the present moment of their existence. This perception includes thoughts, senses, emotions, behaviors, and the environment [33]. Mindfulness can eliminate work stress, thereby enhancing employees' resilience and well-being [34]. Meanwhile, Kabat-Zinn focuses on the three core connotations of awareness or mindfulness, including (i) on purpose; (ii) in the present moment; (iii) non-judgmentally, which are currently cited by academic circles as representative arguments [35].

*2.2. Self-Compassion*

Self-compassion, which originates from the Buddhist concept of compassion, has been translated into self-mercy and self-benevolence. "Kindness" is to treat all sentient beings with respect, while "sadness" is to cause or feel their suffering. Self-compassion refers to the attitude of tolerance and understanding towards oneself when one experiences misery and failure. Self-compassion is knowing oneself and giving kindness to oneself [21]. It is inwardly developing sympathetic psychological functions, namely the sensitivity to miserable experiences and the deep desire to relieve misery. Compassion is a feeling that can be perceived through experience. Meanwhile, the concept of self-compassion put forward includes three main components: self-benevolence, human consensus, and mindfulness. These components interact to form compatible psychological effects.

2.2.1. Self-Benevolence

Self-benevolence refers to caring for and understanding ourselves rather than giving harsh self-criticisms and self-accusations. When we make mistakes or fail, we often condemn ourselves, which we are unlikely to do to our good friends or strangers. When our friends encounter uncontrollable situations, such as unemployment or car accidents, it is easy for us to sympathize with them. However, we find it hard to sympathize with ourselves when we fall into the same situation [21]. Nevertheless, we should not continue punishing ourselves for not being "good enough," but we should admit that we are doing our best [36]. Therefore, our hearts are still gentle and encouraging, not harsh and rash. When the external environment is unbearable, there will be a warm feeling and a desire to improve the misery in our hearts. Self-benevolence will enable us to support and understand ourselves.

2.2.2. Human Consensus

In addition to the discussion of the good and evil of human nature, human nature is also the common experience of individual human beings, which is presented through behaviors. Although we all feel that life is full of struggles and failures, we still feel isolated. Having tunnel vision makes us feel lonely. Neff held that the human consensus accepts that there are imperfections in human nature; all people experience failures and mistakes, not all their wishes will come true [22].

2.2.3. Mindfulness

Mindfulness comes from the Buddhist practice, which is related to being in harmony with the world [37]. It is also a way of perceiving the present in a clear, emphatic, and balanced manner, which neither neglects the self nor adheres to the self or the hardships of life; rather, it is a way to experience the present without critical psychology, escaping psychology, or forcing psychology [38].

Why is mindfulness an important component of self-compassion? We must be willing to face and experience our pain and emotions and embrace ourselves with compassion. Neff noted that mindfulness is concerned with pressure and misery. If we can perceive our negative thoughts and feelings with mindfulness, we will not agree with our self-loathing.

As long as we can collate our negative self-concept and our actual situation, it can help us eliminate some of the stories we tell ourselves about our inadequacy and worthlessness [39].

Many people fail to perceive obvious misery, especially when encountering life challenges. They often become trapped and lose their awareness, unable to consider their current difficulties [38]. Mindfulness practice can effectively enhance self-sympathy, cultivate compassion and care for others, reduce self-blame, and help calm down the process of over-identification. Compassion is a single experience made up of the interaction of self-kindness, human consensus, and mindfulness. Although the three components of self-compassion are different concepts, they also overlap and are correlated.

The differences between self-compassion and the two common concepts are described as follows:

i.　Self-compassion is not self-pity.

　　(i)　Individuals are immersed in their own problems when they feel self-pity. Often, they forget that others have the same problems, making them ignore their connection with others and think that they are alone in their suffering.

　　(ii)　Self-compassion enables us to see our and others' relevant experiences, while self-pity is self-centered and exaggerates the degree of personal suffering. Thus, self-compassion is to let us see our and others' relevant experiences without distortion. Therefore, self-compassion and self-pity are completely different [22].

ii.　Self-compassion is not self-esteem.

　　(i)　Self-esteem refers to the degree to which we positively evaluate ourselves. We like or value ourselves usually based on comparison with others [22].

　　(ii)　However, self-compassion is not based on positive judgment or evaluation but inner self-compassion. Self-compassion is being connected with others rather than separated, which means that one does not have to be "better" than others to feel good about oneself [22].

### 2.2.4. Subjective Well-Being

Well-being is the evaluation of the degree of satisfaction with life and the intensity of positive and negative emotions [2]. The authors of [40] hold that well-being consists of material possessions, the attribution of interpersonal relationships, and self-realization. On the other hand, it is thought that well-being is not completely equal to simple happiness, and the generating factors of well-being include personal growth, self-acceptance, self-esteem satisfaction, positive life goals, good social relationships, autonomy, potential development, and many other factors [41]. Research on well-being can be divided into four stages:

i.　Moral evaluation: Many philosophers believe that when individuals meet standard moral requirements, they will have a sense of well-being [5,42].

ii.　External objective evaluation: Sociologists measure well-being with the personal economy (such as income) or positive emotions as indicators. Research shows that so-called well-being comes from having more positive emotions in life. For example, when income is high and there are many positive emotions, well-being is naturally higher. However, these are all determined by the observer's values, and the inner perception of the individual may not be as the same as that of the observer. Therefore, this is called "objective well-being".

iii.　Emphasis on "perceived" level: Even in the same scenario, everyone may have different feelings, which objective standards of external observation cannot measure. Therefore, in recent years, studies tend to hold that well-being is a kind of personal subjective experience, which comes from the overall feeling of human beings after evaluating previous parts of their lives. Therefore, this is called "traditional subjective well-being".

iv.   Equal attention to "emotion," "perception," and "mental health": Studies have found that emotion is a factor that cannot be ignored for well-being. Authors put forward the concept of "psychological well-being," which means that individuals devote themselves wholeheartedly to activities, achieve their potential, and achieve a pleasant experience of self-realization or, in activities, realize that their life or heart has been satisfied [41,43]. The concept of "social well-being" and the integrated definitions of emotional well-being, psychological well-being, and social well-being are collectively called "subjective well-being" or "emotional well-being" [44].

The concept of well-being has changed from the initial objective external evaluation standard to personal subjective emotion and perception. Most theories about the definition of well-being contain three aspects, namely perception, emotion, and mental health. Therefore, the consensus in the research field of subjective well-being is that these three factors should be taken into account simultaneously. Therefore, in this study, to understand the well-being of employees at work, psychological well-being, emotional well-being, and social well-being were integrated. In addition, subjective well-being was focused on from the perspective of personal perception and subjective feelings.

### 2.3. Other Variables Affected

### 2.3.1. Workplace Friendship

Companionship and trusted friendship between colleagues can maintain the relationship in the workplace and generate satisfaction. In other words, workplace friendships can enable colleagues with the same workplace and working experience to discuss workplace-related topics [45]. This interpersonal relationship includes mutual commitment, trust, information, value, and fun that can be shared in work and life. Meanwhile, a considerable degree of social support is provided [46]. This is not an office romance, nor is it exclusive [47].

According to the literature, workplace friendship can increase mutual support and information-sharing [48], promote career development [49], improve group work performance and organizational effectiveness [50], enable colleagues in the same workplace to have private informal interaction [51], and even promote job satisfaction [52].

### 2.3.2. Social Support

From the perspective of social exchange, Caplan [53] put forward that social support refers to the fact that others help individuals with psychological resources and emotional support in stressful scenarios through the interactive process. Because of this, individuals can increase their capacity to adapt to the environment and improve their control over stress. Social support allows individuals to seek support from their own social networks to reduce the negative impacts of stress [54]. A study mentioned that social support is the assistance or response provided by important others, such as family, friends, neighbors, or colleagues [55]. Some scholars put forward a scale for social support to measure the three main sources of social support, namely family, friends, and important others, and each source is measured for social support with four questions [56]. Social support can also serve informational, instrumental, and appraisal functions [57].

Some studies have proposed that there is a positive relationship between social support and subjective well-being [58], and divided social support into two categories: the main effect and the buffering effect. By satisfying individual needs, an individual's physical and mental health, subjective well-being, emotional relief, and stress release will all have considerable influences [59].

### 2.4. Hypothesis

### 2.4.1. Relationship between Mindfulness and Subjective Well-Being

According to the conceptual framework presented as below Figure 1.

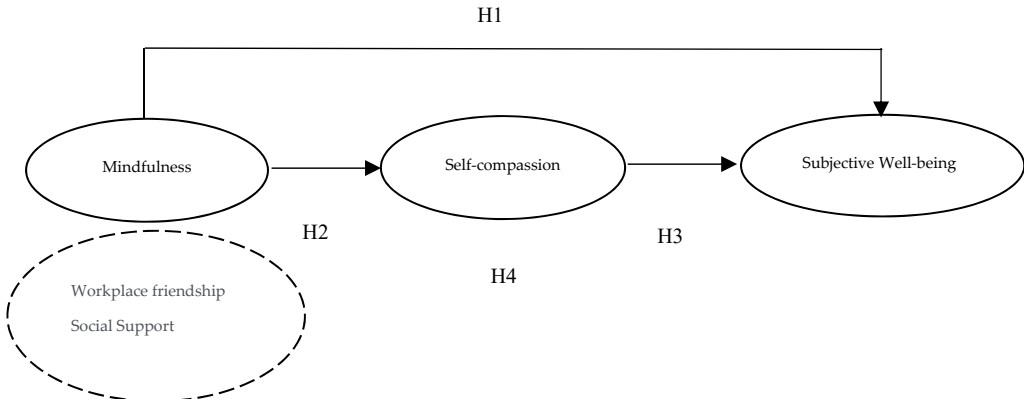

**Figure 1.** Research framework.

**Hypothesis 1 (H1).** *There is a positive relationship between mindfulness and subjective well-being.*

Mindfulness can significantly improve individuals' stress, anxiety, depression, and other negative emotions and generate a relatively high subjective well-being for individuals [60]. Anxiety can reach a state of catatonic ataraxia, a sort of infernal "no exit," transgressing the cycle of anxiety [61]. It can also enhance individuals' awareness of their present experiences. When individuals devote themselves wholeheartedly to activities, they can achieve psychological satisfaction and self-realization through self-regulating behaviors, thereby gaining subjective well-being [41]. According to the literature, the greatest effect of mindfulness lies in relieving individual stress and psychological distress and then obtaining subjective well-being. Based on the above discussion, it is found that mindfulness makes people live in the present and meet individual autonomy and basic psychological needs through self-regulation, helping individuals improve their subjective well-being.

2.4.2. Relationship between Mindfulness and Self-Compassion

**Hypothesis 2 (H2).** *There is a positive relationship between mindfulness and self-compassion.*

According to Neff, the concept of self-compassion contains three main components: self-benevolence, human consensus, and mindfulness [21]. These components interact with each other to form compatible psychological effects. Mindfulness is an important component of self-compassion.

2.4.3. Relationship between Self-Compassion and Subjective Well-Being

**Hypothesis 3 (H3).** *There is a positive relationship between self-compassion and subjective well-being.*

Neff found that self-compassion is positively correlated with positive psychological traits, such as wisdom, happiness, optimism, extroversion, and responsibility. This finding shows that self-compassion can affect individuals' attitudes towards life and help them improve their subjective well-being [21]. Researchers believe that self-compassion could make individuals feel loved and connected, improving their subjective well-being [62]. Furthermore, self-compassion predicts subjective well-being more strongly than social support [63]. With improved self-compassion, individuals reduce self-criticism, depression, and anxiety, giving them better subjective well-being [64].

2.4.4. Relationship among Mindfulness, Self-Compassion, and Subjective Well-Being

**Hypothesis 4 (H4).** *Self-compassion has a mediating effect on the relationship between mindfulness and subjective well-being.*

Neff put forward the concept of self-compassion, which contains three main components: self-benevolence, human consensus, and mindfulness [21]. A study on social work

practitioners confirmed that mindfulness affects overall subjective well-being and also pointed out that mindfulness is an important indicator of life satisfaction [65]. At the emotional and cognitive levels, self-compassion also has a significant mediating effect between psychological well-being and mindfulness [66]. When we have to face and experience our own pain and emotions, we should embrace ourselves with compassion. Based on the above discussion, it is found that mindfulness makes people live in the present and meet individual autonomy and basic psychological needs through self-regulation, helping them improve their subjective well-being.

## 3. Materials and Methods

### 3.1. Measurements

3.1.1. Mindfulness

The operational definition of mindfulness in the FFMQ scale is as follows: (i) attention and action awareness, (ii) observation, (iii) non-judgment, (iv) non-reactiveness to inner experience, and (v) description [67]. In order to make the semantics conform to Taiwanese usage, the Taiwanese version of the five dimensions of mindfulness questionnaire (T-FFMQ) has been re-compiled with 15 items. Items were rated on a 7-point Likert-type scale ranging from 1 (Strongly Disagree) to 7 (Strongly Agree).

**Example 1.** *When I take a shower or bath, I stay alert to the sensations of water on my body.*

3.1.2. Self-Compassion

Self-compassion refers to an individual's kindness, understanding, and less strict self-criticism of themselves when they experience misery or failure. This study used the self-compassion scale compiled by Neff, with 26 items [21]. Items were rated on a 7-point Likert-type scale ranging from 1 (Strongly Disagree) to 7 (Strongly Agree).

**Example 2.** *When I'm feeling down, I tend to obsess and fixate on everything that's wrong.*

3.1.3. Subjective Well-Being

Subjective well-being holds that individuals can participate in social roles to affirm the needs of self-concept and value, self-satisfaction, interpersonal interaction, and social support, making them feel a sense of accomplishment and well-being. This study used the measurement tools defined by Argyle, Martin, and Crossland, with 13 items [6]. Items were rated on a 7-point Likert-type scale ranging from 1 (Strongly Disagree) to 7 (Strongly Agree).

**Example 3.** *I am intensely interested in other people.*

3.1.4. Workplace Friendship

Workplace friendship refers to the degree to which individuals feel the friendly and close interpersonal relationship between themselves and other employees in the workplace. This study used the friendship universality scale developed by Nielsen et al. to measure the depth of workplace friendship among respondents, with 12 items [46]. Items were rated on a 7-point Likert-type scale ranging from 1 (Strongly Disagree) to 7 (Strongly Agree).

**Example 4.** *I have the opportunity to get to know my coworkers.*

3.1.5. Social Support

Social support may come from family, friends, relatives, work partners, and others with significant influences. In this study, the measurement was adapted from Zimet et al. There are three main sources of social support that people need: family, friends, and important others. Therefore, this study used the multi-dimensional scale of perceived social support (MSPSS) compiled by the aforementioned authors, with 12 items [56]. Items were rated on a 7-point Likert-type scale ranging from 1 (Strongly Disagree) to 7 (Strongly Agree).

**Example 5.** *There is a special person who is around when I am in need.*

*3.2. Sample*

In this study, the employees of Carrefour, an international business, were taken as the subjects, and the questionnaire was distributed using Google Forms. The supervisors and employees at Carrefour completed it with mobile phones using the Line communication software. The time period lasted for two months, from January 2021 to February 2021. In terms of measurement tools, questionnaire design and quantitative statistical analysis were used, and the sample selection method was intentional sampling. This study is quantitative. In order to verify the hypotheses of this study, the data analysis methods included reliability and validity analysis, correlation analysis, hierarchical regression analysis, and mediating effect analysis of the bootstrap method. Furthermore, statistical software SPSS22.0 was used as the data analysis tool.

## 4. Results

*4.1. Common Method Bias*

The collected data were tested for common method bias using Harman's single-factor test. Exploratory factor analysis results show that the amount of variance explained by the first common factor was 27.44%, which is below the 40% threshold, showing that the data did not suffer from serious common method bias.

*4.2. Reliability Analysis*

The reliability of this research scale was based on Cronbach's $\alpha$ coefficient as the internal indicator, in which mindfulness was 0.846, self-compassion was 0.907, subjective well-being was 0.858, workplace friendship was 0.832, and social support was 0.828. The display consistency was greater than the standard value of 0.7 proposed by Hair, Tatham, Anderson, and Black, which showed good internal consistency—that is, good reliability. The variables are shown in Table 1.

**Table 1.** Reliability Analysis.

| Variables | Items | Cronbach's $\alpha$ |
|---|---|---|
| Mindfulness | 15 | 0.846 |
| Self-Compassion | 26 | 0.907 |
| Subjective Well-Being | 13 | 0.858 |
| Workplace Friendship | 12 | 0.832 |
| Social Support | 12 | 0.828 |

*4.3. Validity Analysis*

This research questionnaire was based on literature discussion on the development theory and quoted scale items used by most scholars. The content was measured using scholars' judgment indicators, deeming the scale with face validity. Furthermore, the questionnaire was revised and predicted by scholars so that it could meet the standard of content validity. Therefore, the scale had considerable face validity and content validity.

*4.4. Correlation Analysis*

Pearson correlation analysis was used to study the correlation degree and significant levels between variables. The results of the correlation analysis of variables in this study are shown in Table 2. There was a significant positive correlation between mindfulness and self-compassion ($r = 0.815$, $p < 0.01$), indicating that those with high mindfulness have higher self-compassion. There was a significant positive correlation between mindfulness and subjective well-being ($r = 0.653$, $p < 0.01$), implying that those with higher mindfulness have higher subjective well-being. There was a significant positive correlation between self-compassion and subjective well-being ($r = 0.747$, $p < 0.01$), indicating that those with high self-compassion have higher subjective well-being. There was a positive correlation between workplace friendship and subjective well-being ($r = 0.599$, $p < 0.01$), reaching a

significant level. There was a positive correlation between social support and subjective well-being (r = 0.579, *p* < 0.01), reaching a significant level.

**Table 2.** Pearson correlation analysis matrix of each variable.

|  | **Mindfulness** | **Self-Compassion** | **Subjective Well-Being** | **Workplace Friendship** | **Social Support** |
|---|---|---|---|---|---|
| Mindfulness | 1 |  |  |  |  |
| Self-Compassion | 0.815 ** | 1 |  |  |  |
| Subjective Well-Being | 0.653 ** | 0.747 ** | 1 |  |  |
| Workplace Friendship | 0.489 ** | 0.603 ** | 0.599 ** | 1 |  |
| Social Support | 0.415 ** | 0.527 ** | 0.579 ** | 0.673 ** | 1 |

Note: ** *p* < 0.01, indicating a significant level.

*4.5. Regression Analysis*

Regression analysis is a statistical model used to explore the functional relationship between reaction quantity and several explanatory variables. Its purpose is to explain the degree of change of each variable. This study used hierarchical regression analysis to test the mediation model. The analysis results are shown in Figure 2.

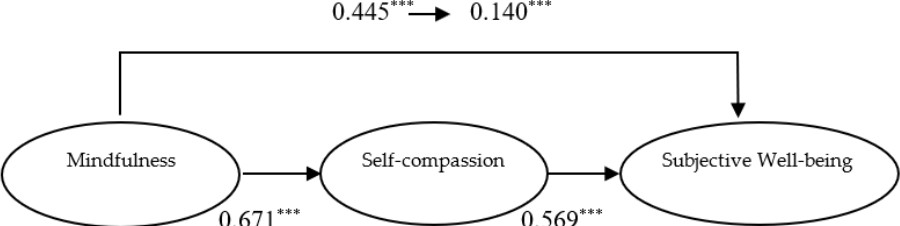

**Figure 2.** Relationships among self-compassion, mindfulness and subjective well-being. Note: *** *p* < 0.001, indicating a significant level.

i.      Model 2 of Table 3 shows the regression analysis of "mindfulness" to "subjective well-being". According to the results, mindfulness was positively and significantly correlated with subjective well-being (F = 268.343, β = 0.445, *p* values all less than 0.001). Therefore, H1 was supported, demonstrating that subjects with higher "mindfulness" have higher "subjective well-being".

ii.     Model 3 of Table 3 shows the regression analysis of mindfulness to self-compassion. According to the results, "mindfulness" was positively and significantly correlated with "self-compassion" (F = 552.686, β = 0.671, *p* values all less than 0.01). Therefore, H2 was supported, indicating that subjects with higher "mindfulness" have higher "self-compassion".

iii.    Model 4 of Table 3 shows the regression analysis of self-compassion to subjective well-being. According to the results, "self-compassion" was positively and significantly correlated with "subjective well-being" (F = 329.666, *p*-value less than 0.001, β = 0.569, *p*-value less than 0.001). Therefore, H3 was supported, demonstrating that subjects with higher "self-compassion" have higher "subjective well-being".

iv.     Model 5 of Table 3 shows the analysis of the mediation effect between mindfulness and subjective well-being by adding self-compassion. According to the results, after "self-compassion" (β = 0.445, *p* < 0.001) was added to the "mindfulness" group (β = 0.140, *p* < 0.001), the positive relationship between "mindfulness" and "subjective well-being" was more significant, and the F value was 258.839 (*p* < 0.001). Therefore, H4 was supported; that is, "self-compassion" has a partial mediating effect on the relationship between "mindfulness" and "subjective well-being".

**Table 3.** Regression analysis results of mindfulness to subjective well-being.

| | Subjective Well-Being | Subjective Well-Being | Self-Compassion | Subjective Well-Being | Subjective Well-Being |
|---|---|---|---|---|---|
| | Model i | Model ii | Model iii | Model iv | Model v |
| | CV→DV | CV + IV →DV | CV + IV →INV | CV + INV →DV | CV + IV + INV→DV |
| Workplace Friendship | 0.382 ** | 0.212 *** | 0.197 *** | 0.124 *** | 0.122 *** |
| Social Support | 0.322 * | 0.252 *** | 0.116 *** | 0.196 *** | 0.199 *** |
| Mindfulness | | 0.445 *** | 0.0671 *** | | 0.140 *** |
| Self-Compassion | | | | 0.569 *** | 0.454 *** |
| $R^2$ | 0.413 | 0.561 | 0.725 | 0.611 | 0.617 |
| $\Delta R^2$ | | 0.148 | | 0.198 | 0.056 |
| F Value | 222.156 *** | 268.343 *** | 552.686 *** | 329.666 *** | 253.839 *** |

Note: * $p < 0.05$; ** $p < 0.01$; *** $p < 0.001$, indicating a significant level.

### 4.6. Mediating Effect Verification with Process Model 4

Table 4 shows the results of the bootstrap resampling with the independent variable "mindfulness," the intermediate variable "self-compassion," and the dependent variable "subjective well-being," which were input into model 4 of process procedure for SPSS release. The indirect effect was 0.4209, while the 95% confidence interval of the indirect effect established by 5000 instances of bootstrap resampling was {0.3435, 0.5704}, which did not include 0. The direct effect was 0.1056, the *p*-value was less than 0.01, which was significant, while the 95% confidence interval of indirect effects established by 5000 instances of bootstrap resampling was {0.0334, 0.1777}, which did not include 0. Therefore, "self-compassion" had a partial mediating effect on "mindfulness" and "subjective well-being".

Table 5 shows that the indirect effect of the intermediate variable "self-compassion" between the independent variable "mindfulness" and the dependent variable "subjective well-being" accounted for 0.7995, which implies that 79.95% of the total effect of "mindfulness" on "subjective well-being" was caused by the indirect effect of "self-compassion".

According to Table 6, the tolerance values of this study were greater than 0.1, and the expansion factors of variance were less than 10. Therefore, there was no collinearity problem in this study.

The verification results of this study are shown in Table 7.

**Table 4.** Process mediation model validation.

| Direct Effect of Mindfulness on Subjective Well-Being | | | | | |
|---|---|---|---|---|---|
| **Effect** | **SE** | **t** | **p** | **LLCI** | **ULCI** |
| 0.1056 | 0.0367 | 20.8734 | 0.0042 ** | 0.0334 | 0.1777 |
| Indirect Effect of Mindfulness on Subjective Well-Being | | | | | |
| | Effect | Boot SE | | BootLLCI | BootULCI |
| Self-Compassion | 0.4209 | 0.0417 | | 0.3435 | 0.5074 |

Note: ** $p < 0.01$, indicating a significant level.

**Table 5.** Process mediation model validation.

| The Ratio of Indirect to Total Effect of Mindfulness on Subjective Well-Being | | | | |
|---|---|---|---|---|
| | **Effect** | **Boot SE** | **LLCI** | **ULCI** |
| Self-Compassion | 0.7995 | 0.0839 | 0.6430 | 0.9719 |

**Table 6.** Collinearity verification between the independent variable and the intermediate variable.

| Construct<br>Item | Constant | Mindfulness | Self-Compassion | Workplace<br>Friendship | Social Support |
|---|---|---|---|---|---|
| Tolerance Value | | 0.336 | 0.274 | 0.462 | 0.523 |
| Expansion Factor | | 0.980 | 3.653 | 2.166 | 1.911 |

**Table 7.** Empirical results of research hypotheses.

| Hypothesis | Content | Verification Result |
|---|---|---|
| H1 | There is a positive relationship between mindfulness and subjective well-being. | Valid |
| H2 | There is a positive relationship between mindfulness and self-compassion. | Valid |
| H3 | There is a positive relationship between self-compassion and subjective well-being. | Valid |
| H4 | Self-compassion has a mediating effect on the relationship between mindfulness and subjective well-being. | Valid |

## 5. Conclusions

i.     There is a positive relationship between mindfulness and subjective well-being.

According to the empirical results, there is a positive relationship between mindfulness and subjective well-being (H1). Mindfulness is found to significantly relieve individuals' stress, anxiety, and depression and generate higher subjective well-being for individuals [60]. Mindfulness can also enhance individuals' awareness of their present experiences and keep an attitude of acceptance and non-judgment of the course of events. When individuals pay attention all the time, they will have intentional self-regulation [24].

ii.     There is a positive relationship between mindfulness and self-compassion.

According to the empirical results, there is a positive relationship between mindfulness and self-compassion (H2). Mindfulness is found to effectively enhance self-compassion, cultivate compassion and care for others, reduce the degree of self-blame, and help quell the process of over-identification. Taking the common experience of human beings as the foundation, coupled with self-compassion, we can adopt a sympathetic attitude towards ourselves and a broader view. Moreover, when we are faced with misery, we will not feel so isolated. As Neff said, accepting mindfulness helps reduce self-criticism and allows us to learn about our common humanity [39].

iii.     There is a positive relationship between self-compassion and subjective well-being.

According to the empirical results, there is a positive relationship between self-compassion and subjective well-being (H3). Research shows that self-compassion is positively correlated with positive psychological traits, such as wisdom, happiness, optimism, extroversion, and responsibility. This assertion indicates that self-compassion can affect individuals' attitudes towards life and help individuals improve their subjective well-being [21]. It is also thought that self-compassion can make individuals feel loved and connected, thereby improving their subjective well-being [62]. With improved self-compassion, individuals reduce self-criticism, depression, and anxiety, giving them more subjective well-being [64].

iv.     Self-compassion has a mediating effect on the relationship between mindfulness and subjective well-being.

According to the empirical results, self-compassion has a mediating effect on the relationship between mindfulness and subjective well-being (H4). Neff noted that as long as we can collate our negative self-concept and actual self-situation, it can help us

eliminate some stories we tell ourselves about our own inadequacy and worthless ness [39]. Mindfulness is concerned with pressure and misery. If we can perceive our negative thoughts and feelings with mindfulness, we will not agree with our self-loathing.

From the above discussion, it can be deduced that mindfulness makes people live in the present, which can be adjusted by self-compassion to meet individual autonomy and basic psychological needs and improve subjective well-being.

## 6. Theoretical Implications and Practical Implications

### 6.1. Theoretical Implications

In this study, self-compassion was taken as the mediating variable to discuss the impact of mindfulness on subjective well-being. According to the literature, the greatest effect of mindfulness lies in relieving individual stress and psychological distress and then obtaining subjective well-being. Mindfulness occurs when people live in the present and meet individual autonomy and basic psychological needs through self-regulation, thus helping to improve subjective well-being. The analysis results have the following theoretical significance.

i.　　There is a positive relationship between mindfulness and subjective well-being. When individuals devote themselves wholeheartedly to activities, they can achieve psychological satisfaction and self-realization through self-regulating behaviors, and then gain subjective well-being [41].

ii.　　There is a positive relationship between mindfulness and self-compassion. Mindfulness can stimulate self-healing power and then improve self-compassion, which is consistent with the three main components put forward by Neff for the concept of self-compassion [21].

iii.　　There is a positive relationship between self-compassion and subjective well-being. Self-compassion can predict subjective well-being much better than social support [63].

iv.　　Self-compassion has a mediating effect on the relationship between mindfulness and subjective well-being. According to the results, subjective well-being is mainly positively affected by mindfulness to improve self-compassion.

### 6.2. Practical Implications

According to the results, subjective well-being can be enhanced by mindfulness, and self-compassion has a mediating effect on the relationship between mindfulness and subjective well-being, which has a practical significance. Additionally, well-being has a good prediction effect on work performance. There are several findings from this study:

i.　　Mindfulness training courses can be provided for employees.

In the mindfulness questionnaire, question 4 got the lowest average, that is, "I think that some of my ideas are not good or abnormal, and I do not think that I should have thought that way". Mindfulness improves mental skills through learning, and this skill can help an individual reduce personal physiological deficiencies and improve personal well-being [29]. For such mental defects that often appear in modern people, mindfulness education can provide a possible way for an individual to face and deal with a negative self-relationship.

ii.　　Self-compassion training courses can be provided for employees.

In the self-compassion questionnaire, question 4 got the lowest average, that is, "When I lose something important, I will assume the failure alone". Neff put forward that self-compassion is to know oneself and generate compassion for oneself [21]. This is an inward developing sympathetic psychological function, namely the sensitivity to miserable experiences and the deep desire to relieve misery. Therefore, when facing misery, one is taken as a subject to be cared for and included in sympathy without isolating oneself so as to show mercy to the misery of oneself and that of others.

iii.　　Employees can be assisted in building a connection with workmates.

In the workplace friendship questionnaire, question 12 got the lowest average, that is, "I do not think that I have one real friend in the workplace". Workplace friendship can improve group performance and increase organization effectiveness [50]. Therefore, employees can be assisted in building connections with workmates to interact well with each other, which can bring about the continuous development of the enterprise.

## 7. Limitations and Suggestions

### 7.1. Limitations

i.     Research Subjects:

Due to the limitation in funds and time, the data were collected by intentional sampling. The sampling objects included employees of Carrefour, an international business. However, in the actual sampling, the familiar Carrefour employees were entrusted to help send out questionnaires. Therefore, when these employees sent out questionnaires, the objects were not random but rather close to them. This way, the questionnaires would affect the subjects.

ii.     Research Method:

In this study, the questionnaire survey method was used. Although the questionnaire format was well-organized and easy to answer, the respondents might have answered incorrectly if they did not know the meanings of the questions. Furthermore, this questionnaire was a closed questionnaire, which could only show the respondents' perception of mindfulness, self-compassion, and subjective well-being, failing to clearly determine the real reason behind the respondents' perceptions. Finally, the results presented in the questionnaire were only the parts that the respondents were willing to show to others, which might not be consistent with their facts.

### 7.2. Suggestions

The sample scope of this study was the Carrefour employees. Future researchers may take employees of other businesses as subjects when distributing questionnaires. Since each business has different organizational culture and leadership styles, whether the relationship between employees' mindfulness and subjective well-being is different or not should be clarified by future researchers.

Finally, future researchers may use different statistical methods to explore other problems of employees in international businesses. For instance, researchers may use an independent sample T-test to explore whether there are obvious differences in the subjective well-being of male and female employees or a one-way ANOVA test to discuss whether there are obvious differences in the subjective well-being of employees with different service years. In this way, the relationship between mindfulness and the subjective well-being of employees in international businesses may be understood from many aspects.

**Author Contributions:** Theoretical model design, F.-H.Y.; data collection, Y.-L.L.; validation: F.-H.Y., S.-L.T.; methodology, F.-H.Y. and S.-L.T.; software, F.-H.Y.; writing—original draft, Y.-L.L.; writing—review & editing, F.-H.Y. and S.-L.T. All authors have read and agreed to the published version of the manuscript.

**Funding:** This research received no external funding.

**Institutional Review Board Statement:** Not applicable.

**Informed Consent Statement:** Not applicable.

**Data Availability Statement:** The data presented in this study are available on request from the corresponding author.

**Acknowledgments:** We would like to thank Meng Li Huang for his kind help and constructive suggestions.

**Conflicts of Interest:** The authors declare no conflict of interest.

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
