# Peer review of "The Relationships among Mindfulness, Self-Compassion, and Subjective Well-Being: The Case of Employees in an International Business"

_sustainability, doi:10.3390/su14095266_

Round 1
Reviewer 1 Report
Dear author(s), I have read your paper and I need to say that it is really interesting. I think it will have good acceptance in the public. You demonstrated good research skills, and knowledge in the area. On the other hand, there are some points that you need to address before the paper could be published.
First, you stated your hypotheses in the material and methods section. Please, go trough the text and place the right hypotheses on the right place in the text. Hypotheses should be developed from the theory, not at the end or in the other sections.
Second, describe with some addition research results the relations in the section 3.1.4, there is only one research that was cited.
Third, we don't see the questions for the variables. Please, present some of them in the text or in the table.
Fourth, what is the response rate? How did you deal with the common method bias?
Fifth, please, make the results more clear. There is al lot of tables, but you have only 4 hypotheses. Use the Process for mediation, and explain that relation in more details.
Sixth, please rewrite practical and theoretical implications and make them sounder.
Author Response
RESPONSE TO THE REVIEWER’ COMMENTS | ||
Item | Reviewers’ Comments/Remarks | adjusted & revised |
1 | First, you stated your hypotheses in the material and methods section. Please, go trough the text and place the right hypotheses on the right place in the text. Hypotheses should be developed from the theory, not at the end or in the other sections | modify field as attacthment |
2 | Second, describe with some addition research results the relations in the section 3.1.4, there is only one research that was cited. | Add references to describe in this section: -Shier M. L., Graham J. R. Mindfulness, Subjective Well-Being, and Social Work: Insight into their Interconnection from Social Work Practitioners 2011, 30, 29-40 -Laurie H. W., Colosimo K. Mindfulness, self-compassion, and happiness in non-meditators: A theoretical and empirical examination 2011. 50. 222-227. |
3 | Third, we don't see the questions for the variables. Please, present some of them in the text or in the table. | Had Added Questionnaire examples for variables. Line 384 392 401 409 418 |
4 | Fourth, what is the response rate? How did you deal with the common method bias? | Add 4.1 4.1. Common Method Bias The collected data were tested for common method bias using Harman’s sin-gle-factor test. Exploratory factor analysis results show that the amount of variance ex-plained by first common factor was 27.44%, which is below the 40% threshold, show-ing that the data did not suffer from serious common method bias. |
5 | Fifth, please, make the results more clear. There is al lot of tables, but you have only 4 hypotheses. Use the Process for mediation, and explain that relation in more details. | Cancelled Table 5 Cancelled Table 6 |
6 | Sixth, please rewrite practical and theoretical implications and make them sounder | Had Added practical and theoretical implications . Line 494-545 |

Reviewer 2 Report
Dear authors:
The topic of the research proposal is very interesting and I congratulate for your efforts. In order to publish the paper, I recommend you some improvements
It will be good to add some papers from different regions:
Lines: 53-55
According to relevant studies, subjective well-being is affected by personality traits, such as being grateful [8], getting more social support [9] [10] [11], good interpersonal relationships [12] [13], family structure, and demographic [14] [15].
- as being grateful - would be good to add these articles https://doi.org/10.3390/su131910826 : Petrovič, F., Murgaš, F., & Králik, R. (2021). Happiness in Czechia during the COVID-19 Pandemic. Sustainability, 13(19), 10826. https://doi.org/10.3390/su131910826
- family structure, and demographic - https://doi.org/10.3390/ijerph19052767
- good interpersonal relationships - would be good to add this article: https://doi.org/10.3390/su131810350
- https://doi.org/10.3390/su14042049
- Kobylarek, A., Błaszczyński, K., Ślósarz, L., Madej, M., Carmo, A., Hlad, Ľ., Králik, R., Akimjak, A., Judák, V., Maturkanič, P., Biryukova, Y., Tokárová, B., Martin, J. G., & Petrikovičová, L. (2022). The Quality of Life among University of the Third Age Students in Poland, Ukraine and Belarus. Sustainability, 14(4), 2049. https://doi.org/10.3390/su14042049
concept of anxiety: line: (250) - https://doi.org/10.15503/jecs20151.20.25
https://doi.org/10.15503/jecs2020.2.224.236
line: 185 connection with doi: 10.21125/iceri.2019.2129
and
Martin, José García, Arturo Rojas Morales, and Roman Králik. 2020. The problem of the ‘individual’ concept in the Kierkegaard’s
journals. European Journal of Science and Theology 16: 39–46.
Good luck !
Author Response
RESPONSE TO THE REVIEWER’ COMMENTS | ||
Item | Reviewers’ Comments/Remarks | adjusted & revised |
1 | Lines: 53-55 According to relevant studies, subjective well-being is affected by personality traits, such as being grateful [8], getting more social support [9] [10] [11], good interpersonal relationships [12] [13], family structure, and demographic [14] [15]. as being grateful - would be good to add these articles https://doi.org/10.3390/su131910826 : Petrovič, F., Murgaš, F., & Králik, R. (2021). Happiness in Czechia during the COVID-19 Pandemic. Sustainability, 13(19), 10826. https://doi.org/10.3390/su131910826 family structure, and demographic - https://doi.org/10.3390/ijerph19052767 good interpersonal relationships - would be good to add this article: https://doi.org/10.3390/su131810350 https://doi.org/10.3390/su14042049 Kobylarek, A., Błaszczyński, K., Ślósarz, L., Madej, M., Carmo, A., Hlad, Ľ., Králik, R., Akimjak, A., Judák, V., Maturkanič, P., Biryukova, Y., Tokárová, B., Martin, J. G., & Petrikovičová, L. (2022). The Quality of Life among University of the Third Age Students in Poland, Ukraine and Belarus. Sustainability, 14(4), 2049. https://doi.org/10.3390/su14042049 |
line 52-56 Add The COVID-19 pandemic in 2020, happiness has not decreased because of the epidemic, and women's happiness has not decreased, the main reason may be affected by other experiences, such as quality of life and place quality. There was also no reduction in women's happiness, which could signal a feminist social shift.[9] line 59 Add family structure, and demographi Credibility and Involvement of Social Media in Education-Recommendations for Mitigating the Negative Effects of the Pandemic among High School Students[20] good interpersonal relationships line 59 How to Keep University Active during COVID-19 Pandemic: Experience from Slovakia[16] line 59 The Quality of Life among University of the Third Age Students in Poland, Ukraine and Belarus[17] |
2 | concept of anxiety: line: (250) - https://doi.org/10.15503/jecs20151.20.25 https://doi.org/10.15503/jecs2020.2.224.236 |
Add reference(The Courage To Be Anxious. Paul Tillich’s Existential Interpretation of Anxiety ) line 252 |
3 | line: 185 connection with doi: 10.21125/iceri.2019.2129 | Add reference(Interpersonal relationships as the basis of student moral formation ) line 185 |
4 | Martin, José García, Arturo Rojas Morales, and Roman Králik. 2020. The problem of the ‘individual’ concept in the Kierkegaard’s journals. European Journal of Science and Theology 16: 39–46. |
line 39-41 Add(The problem of the ‘individual’ concept in the Kierkegaard’s journals.) It with respect to the individual: “I have nothing to do with the world. I have to do with the Individual, each Individual, or to each Individual[4] |

Reviewer 3 Report
This is a review of the manuscript titled "The Relationships among Mindfulness, Self-Compassion, and Subjective Well-Being: The case of employees in an international business". This study aimed at investigating the relationships among mindfulness, self-compassion, and subjective well-being among employees. The results showed that mindfulness was positively related to subjective well-being and, mindfulness was positively related to self-compassion, and self-compassion was positively related to subjective well-being. It was also found that self-compassed partially mediated the relationship etween mindfulness and subjective well-being. This manuscript could be improved if the following concerns are addressed:
- I suggest the authors not to present the findings of the study in point form in the Abstract.
- The authors list various definitions of mindfulness proposed by different scholars (p. 2). Some definitions are similar. I suggest the authors integrate them.
- It is stated that mindfulness consists of five dimensions according to the research on the Five Dimension Mindfulness Questionnaire (FFMQ) (p. 3). Please briefly explain the meaning of each dimension.
- It is stated that social support involves the provision of psychological resources and emotional support (p. 5). I suggest the authors add that social support can also serve informational, instrumental, and appraisal functions (Ng et al., 2017).
Ng, T. K., Wang, K. W. C., & Chan, W. (2017). Acculturation and cross-cultural adaptation: The moderating role of social support. International Journal of Intercultural Relations, 59, 19-30. https://doi.org/10.1016/j.ijintrel.2017.04.012
- The Hypotheses section (p. 6) should be presented before the Materials and Methods section instead of being included as its subsection.
- For the measurements (p. 7), please provide the response format for each scale (e.g., 5-point scale)
- The KMO and Bartlett’s test are normally used to examine whether the data are suitable for factor analysis. They are not necessary if no factor analysis is conducted.
- The authors state that, "Model 3 of Table 4 shows the regression analysis of mindfulness to psychological security." (p. 10). It should be self-compassion instead of psychological security.
- The authors state that, "Model 5 of Table 4 shows the analysis of the interaction effect between mindfulness and subjective well-being by adding self-compassion." (p. 10) This is not an interaction (moderating) effect.
- Please clarify whether workplace friendship and social support were included as covariates in the mediation analysis.
Author Response
RESPONSE TO THE REVIEWER’ COMMENTS | ||
Item | Reviewers’ Comments/Remarks | Authors’ Comments |
This is a review of the manuscript titled "The Relationships among Mindfulness, Self-Compassion, and Subjective Well-Being: The case of employees in an international business". This study aimed at investigating the relationships among mindfulness, self-compassion, and subjective well-being among employees. The results showed that mindfulness was positively related to subjective well-being and, mindfulness was positively related to self-compassion, and self-compassion was positively related to subjective well-being. It was also found that self-compassed partially mediated the relationship etween mindfulness and subjective well-being. | Revised title: The Relationships among Mindfulness, Self-Compassion, and Subjective Well-Being Revised Abstract: This study aimed at investigating the relationships among mindfulness, self-compassion, and subjective well-being among employees. The questionnaire research method was used to collect data in this study, and the subjects included employees of Carrefour, an international business in Taiwan. A total of 629 valid questionnaires were used to evaluate the overall structure and analyze the mediating effect with the SPSS 21.0 statistical software. The results showed that mindfulness was positively related to subjective well-being and, mindfulness was positively related to self-compassion, and self-compassion was positively related to subjective well-being. It was also found that self-compassed partially medi-ated the relationship etween mindfulness and subjective well-being. |
|
1 | I suggest the authors not to present the findings of the study in point form in the Abstract | |
2 | The authors list various definitions of mindfulness proposed by different scholars (p. 2). Some definitions are similar. I suggest the authors integrate them. | Had Revised Much literature defined mindfulness as the present experience through conscious awareness without evaluation defind mindfulness as the awareness of one's sur-roundings, presenting the experience of the moment [29]. Furthermore, mindful-ness emphasizes dealing with things in life intelligently and avoiding blind and in-stinctive reactions [30]. then facing the present expe-rience with curiosity, openness, and acceptance [31] |
3 | It is stated that mindfulness consists of five dimensions according to the research on the Five Dimension Mindfulness Questionnaire (FFMQ) (p. 3). Please briefly explain the meaning of each dimension | Had Revised Mindfulness is a multi-dimensioned concept. Through exploratory factor analysis, five main mindfulness characteristics were found, namely (i) attention/action aware-ness, I find myself doing things without paying attention, (ii) observation, I notice the smells and aromas of things, (iii) non-judgment, I think some of my emotions are bad or inappropriate and I should not feel them, (iv) non-reactive to inner experience, I perceive my feelings and emotions without having to react to them and (v) description, I am good at finding words to describe my feelings, establishing the five dimension mindfulness questionnaire (FFMQ) [32] |
4 | It is stated that social support involves the provision of psychological resources and emotional support (p. 5). I suggest the authors add that social support can also serve informational, instrumental, and appraisal functions (Ng et al., 2017). | Had Revised line 239-240 The social support can also serve informational, instrumental, and appraisal func-tions[57]. |
5 | The Hypotheses section (p. 6) should be presented before the Materials and Methods section instead of being included as its subsection | Had revised Line 248、 263、270、283 |
6 | For the measurements (p. 7), please provide the response format for each scale (e.g., 5-point scale) | Had Revised All Questionnaire scale Items are rated on a 7-point Likert-type scale ranging from 1 (Strongly Disagree) to 7 (Strongly Agree) |
7 | The KMO and Bartlett’s test are normally used to examine whether the data are suitable for factor analysis. They are not necessary if no factor analysis is conducted | 4.2. Validity Analysis This research questionnaire was based on literature discussion on the development theory and quoted scale items used by most scholars. The content was measured by scholars’ judgment in-dicators, deeming the scale with face validity. Further, the questionnaire had been revised and predicted by scholars so that it could meet the standard of content validity. Therefore, the scale had considerable face validity and content validity. Cancelled Table 2 |
8 | The authors state that, "Model 3 of Table 4 shows the regression analysis of mindfulness to psychological security." (p. 10). It should be self-compassion instead of psychological security | The Model 3 of Table 3 shows CV(Workplace friendship & Social Support)+IV(Mindfulness)→INV(self compassion),please to re-examined. |
9 | The authors state that, "Model 5 of Table 4 shows the analysis of the interaction effect between mindfulness and subjective well-being by adding self-compassion." (p. 10) This is not an interaction (moderating) effect. | Had Revised Model 5 of Table 3 shows the analysis of the mediation effect between mindfulness and subjective well-being by adding self-compassion. |
10 | Please clarify whether workplace friendship and social support were included as covariates in the mediation analysis. | Social support and Workplace friendship are control variables, mainly to avoid the effect of Mindfulness on Subjective well-being |

Round 2
Reviewer 1 Report
The paper now is accepted for publication.
Author Response
The paper has revised as attachment

Reviewer 3 Report
The authors have addressed most of the issues adequately and the manuscript has been improved. There is one remaining minor issue. The authors state that, "Model 3 of Table 3 shows the regression analysis of mindfulness to psychological security." (p. 10). It should be "mindfulness to self-compassion " instead of "mindfulness to psychological security".
Author Response
RESPONSE TO THE REVIEWER’ COMMENTS | ||
Item | Reviewers’ Comments/Remarks | Authors’ Comments |
The authors have addressed most of the issues adequately and the manuscript has been improved. There is one remaining minor issue. The authors state that, "Model 3 of Table 3 shows the regression analysis of mindfulness to psychological security." (p. 10). It should be "mindfulness to self-compassion " instead of "mindfulness to psychological security". | Revised ii. Model 3 of Table 3 shows the regression analysis of mindfulness to self-compassion. (p10) |
